# Irinotecan (CPT-11) Canonical Anti-Cancer Drug Can also Modulate Antiviral and Pro-Inflammatory Responses of Primary Human Synovial Fibroblasts

**DOI:** 10.3390/cells10061431

**Published:** 2021-06-08

**Authors:** Anthony Dobi, Philippe Gasque, Pascale Guiraud, Jimmy Selambarom

**Affiliations:** 1Unité de Recherche en Pharmaco-Immunologie (UR-EPI), Université et CHU de La Réunion (Site Félix Guyon), 97400 Saint-Denis, France; philippe.gasque@univ-reunion.fr (P.G.); pascale.guiraud@univ-reunion.fr (P.G.); jimmy.selambarom@univ-reunion.fr (J.S.); 2Laboratoire d’Immunologie Clinique et Expérimentale de la Zone Océan Indien (LICE-OI), Pôle de Biologie, CHU de La Réunion (Site Félix Guyon), 97400 Saint-Denis, France

**Keywords:** irinotecan, arthritis, alphavirus infection, human synovial fibroblasts, antiviral response, pro-inflammatory response

## Abstract

Alphaviruses are a group of arboviruses that generate chronic inflammatory rheumatisms in humans. Currently, no approved vaccines or antiviral therapies are available to prevent or treat alphavirus-induced diseases. The aim of this study was to evaluate the repositioning of the anti-cancer molecule irinotecan as a potential modulator of the antiviral and inflammatory responses of primary human synovial fibroblasts (HSF), the main stromal cells of the joint synovium. HSF were exposed to O’nyong-nyong virus (ONNV) and polyinosinic-polycytidylic acid (PIC) to mimic, respectively, acute and chronic infectious settings. The cytokine IL-1β was used as a major pro-inflammatory cytokine to stimulate HSF. Quantitative RT-PCR analysis revealed that irinotecan at 15 µM was able to amplify the antiviral response (i.e., interferon-stimulated gene expression) of HSF exposed to PIC and reduce the expression of pro-inflammatory genes (CXCL8, IL-6 and COX-2) upon IL-1β treatment. These results were associated with the regulation of the expression of several genes, including those encoding for STAT1, STAT2, p53 and NF-κB. Irinotecan did not modulate these responses in both untreated cells and cells stimulated with ONNV. This suggests that this drug could be therapeutically useful for the treatment of chronic and severe (rather than acute) arthritis due to viruses.

## 1. Introduction

Infectious diseases are a well-recognized cause of acute arthralgia and arthritis. There is currently a lack of accurate data on the incidence and prevalence of viral arthritis worldwide. Studies performed on patients affected by acute arthritis have suggested a viral origin in about 1% of cases [1]. Among arthritogenic viruses, the genus *Alphavirus*, a member of the family *Togaviridae*, has been reported to generate epidemics in most parts of the world, sometimes with a severe impact on human health [2]. Alphaviruses are enveloped positive-sense RNA viruses transmitted by bites of infected mosquitoes and are importantly involved in inflammatory diseases, including arthritis and encephalitis [3]. “Old World” alphaviruses, of which Chikungunya virus (CHIKV) is the most well-known, are related to rheumatic diseases in humans that can turn into a chronic form. Clinical manifestations of chronic arthralgia and arthritis after alphaviral infection range from a restriction of movements with persistence of swelling and pain located in joints to a debilitating illness [4,5]. 

The production of type I interferon (IFN-α and IFN-β) and, subsequently, IFN-stimulated genes (ISGs) by virus-infected cells is a hallmark of the innate immunity, allowing one to control alphaviral replication. However, inflammatory mediators, such as CCL2 and CXCL8, play a central role in disease progression towards a chronic phase, particularly in the process of immune cells’ recruitment in the synovial fluid of inflamed joints [6,7]. Although macrophages are thought to be key effectors of the pathology through the release of pro-inflammatory cytokines, chemokines and matrix metalloproteinases, synovial fibroblasts are also important contributors of inflammation and joint damage. For instance, it has been demonstrated that human synovial fibroblasts (HSF) infected by CHIKV secrete mediators (RANKL, IL-6, IL-8 and CCL2) that are responsible for the recruitment of phagocytes and their differentiation into osteoclast-like cells involved in bone erosion [8]. In this context, the inhibition of virus replication, as well as pro-inflammatory processes related to synovial fibroblasts are of therapeutic interest.

Alkaloids, a class of nitrogen-containing organic compounds, are secondary metabolites of plants that are recognized to inhibit the replication of several viruses, particularly RNA viruses, including CHIKV [9], influenza virus [10], dengue virus [11] and porcine epidemic diarrhea virus [12]. Among alkaloids, water-insoluble camptothecin (CPT) and its derivatives, described as inhibitors of DNA topoisomerase I and traditionally used as anti-cancer drugs, have been shown to affect the replication of many viruses, including DNA viruses [13], influenza virus [14] and HIV-1 [15]. In addition, a genome-wide transcriptional analysis of human fibroblasts treated with CPT showed that the molecule was able to regulate the expression of a wide range of genes, including proto-oncogenes, p53 target genes, pro- and anti-apoptotic genes [16].

Hence, in the present study, our aim was to develop an in vitro model to ascertain whether or not CPT may modulate the antiviral and pro-inflammatory gene expression of HSF in the context of alphaviral infection and viral persistence in the joints of patients affected by chronic inflammatory rheumatisms. We used the camptothecin-derived soluble drug irinotecan (CPT-11) that was approved for the treatment of cancer in 1994 and that remains currently a major anti-cancer drug worldwide [17]. Primary HSF were exposed to O’nyong-nyong virus (ONNV), a virus closely related to CHIKV, with which it shares 89% genetic sequence homology [18]. Polyinosinic-polycytidylic acid (PIC), a synthetic analogue of viral double-stranded RNA, and the pro-inflammatory cytokine IL-1β were also applied to cells. Gene expression analysis revealed that CPT-11 can modulate both the antiviral and pro-inflammatory responses of HSF.

## 2. Materials and Methods

### 2.1. Cells and Reagents

Primary HSF were obtained from ScienCell Research Laboratories (ScienCell, 4700; Clinisciences, Carlsbad, CA, USA). Cells were cultured in Modified Eagle’s Medium (MEM eagle, PAN Biotech P0408500, Aidenbach, Germany) supplemented with 10% heat-inactivated fetal bovine serum (FBS, PAN Biotech, 3302 P290907, Aidenbach, Germany), 2 mM L-glutamine (Biochrom AG, K0282, Berlin, Germany), 0.1 mg/mL penicillin-streptomycin (PAN Biotech, P0607100), 1 mM sodium pyruvate (PAN Biotech, P0443100) and 0.5 µg/mL amphotericin B (PAN Biotech, P0601001).

ONNV was provided by the CNR (Centre national de référence) Arbovirus of Marseille (France). Polyinosinic-polycytidylic acid (PIC) was purchased from Amersham Biosciences, Little Chalfont, UK (catalog number: 27-4732-01) and recombinant human IL-1β and TNF-α from Peprotech, Rocky Hill, CT, USA (catalog number: 200-01B and 300-01A, respectively). Irinotecan hydrochloride trihydrate was obtained from “medac Gesellschaft für klinische Spezialpräparate mbH” (Wedel, Germany).

### 2.2. Cell Culture and Treatments

HSF were cultured in 6-well plates or 96-well plates. Cells were maintained at 37 °C in a humidified atmosphere containing 5% CO_2_. The medium was replaced twice a week and cells were treated at about 80% confluence. All treatments were performed for 6 and 24 h. HSF were infected with ONNV at a multiplicity of infection (MOI) of 0.1 and 1 and under biosafety level 2 practices. After the infection time, tissue culture plates were inactivated with ultraviolet light for 5 min. PIC was used at 1 µg/mL, IL-1β at 10 ng/mL and TNF-α at 50 ng/mL. Cells were co-treated (or not) with CPT-11 at 15 µM.

### 2.3. MTT Assay

MTT (3-(4,5-Dimethylthiazol-2-yl)-2,5-Diphenyltetrazolium Bromide) assay was performed to investigate the influence of ONNV, PIC, IL-1β and CPT-11 on HSF viability. After 24 h of treatment in 96-well plate, MTT was added to each well at a final concentration of 0.5 mg/mL, followed by 3 h of incubation at 37 °C. After discarding the medium, dimethylsulfoxide was added to each well (100 µL) and the plate was shaken for 15 min to solubilize formazan crystals. Absorbance was read at 570 nm. Results were expressed as the percentage of control (untreated cells).

### 2.4. Cytotoxicity Assay (LDH Assay)

Cytotoxicity was assessed with the CytoTox 96^®^ Non-Radioactive Cytotoxicity Assay (PROMEGA, Madison, USA, catalog number: G1780), which is a colorimetric assay allowing for the detection of lactate dehydrogenase (LDH) released from damaged cells, in supernatants. Experiments were carried out in sextuplicate in 96-well plates. The percentage of cellular injury was calculated using the following formula: percent cytotoxicity = 100 × experimental LDH release/maximum LDH release. Maximum LDH release was determined after 24 h of cell treatment with 0.5% triton.

### 2.5. Reverse Transcription Quantitative Real-Time Polymerase Chain Reaction (RT-qPCR)

Total RNA extraction was performed using the RNeasy Plus Mini Kit (Qiagen, Hilden, Germany, catalog number: 74136). HSF were cultured and treated in 6-well plates. After treatments, supernatants were collected and stored at −80 °C. Cells were washed with 1X PBS and lysed with 350 µL RLT lysis buffer. Cell lysates were stored at −80 °C until RNA extraction, according to the manufacturer’s instructions. Total RNA was eluted in a final volume of 50 µL RNase free water and used for RT-qPCR experiments, otherwise stored at −80 °C. RT-qPCR was carried out in triplicate using the One-Step TB Green PrimeScript™ RT-PCR Kit (Takara Bio Inc., Kusatsu, Japan, catalog number: RR066A) in a final volume of 5 μL containing 1 μL of total RNA (or RNase free water as a blank non-template control), 2.7 μL of enzyme mix and 1.3 μL of primers mix at a final concentration of 250 nM. The specific primers used are listed in Table 1. The RT-qPCR program was executed with the QuantStudio 3 Real-Time PCR System with the following steps: a reverse transcription at 42 °C for 5 min and 40 cycles comprising a denaturation step at 95 °C for 5 s, an annealing step at 58 °C for 15 s and an extension step at 72 °C for 15 s. Fluorescence data were collected at 520 nm. PCR reactions were validated by analyzing the melting curve of each pair of primers with the QuantStudio Design and Analysis Software v1.5.1 (ThermoFisher Scientific, Waltham, MA, USA). Relative gene expression was calculated using GAPDH as a housekeeping gene and the fold change versus control (set to 1) was calculated for each gene. Of note, when control values were equal to zero (in the case of viral genes’ amplification in uninfected cells), the 2^−∆Ct^ values were reported for other conditions [19].

### 2.6. Enzyme-Linked Immunosorbent Assay (ELISA)

Cytokine and chemokine concentrations in supernatants of HSF were measured using commercially available ELISA kits for CXCL8 (Peprotech, Rocky Hill, CT, USA, catalog number: 900-T18) and IL-6 (eBioscience, ThermoFisher Scientific, Waltham, MA, USA, catalog number: 88-7066-88), according to the manufacturer’s instructions.

### 2.7. Western Blot Analysis

Cultured HSF were washed with PBS and cells were scraped off the plate. Protein extraction was performed using 1% Triton X-100 lysis buffer (150 mM sodium chloride, 50 mM Tris pH 7.5) supplemented with a cocktail of protease inhibitors (ThermoFischer Scientific, Waltham, MA, USA, catalog number: A32961). Equal amounts of cell lysate protein (10 µg), determined by Bradford assay, were mixed with 1X Laemmli buffer and then loaded on NuPAGE 4-12% Bis-Tris Gel (Invitrogen, ThermoFisher Scientific, Waltham, MA, USA, catalog number: NP0336BOX) for SDS-PAGE electrophoresis (50 mA per gel, for one hour and thirty minutes). Separated proteins were transferred to a 0.45 µm nitrocellulose membrane (Amersham Biosciences, Little Chalfont, UK, catalog number: RPN303E) at 50 mA, for one hour and fifteen minutes. The membrane was then blocked with 5% non-fat dry milk containing 0.1% Tween 20 and incubated overnight at 4 °C with primary antibodies against COX-2 (Rabbit anti-COX-2, Cell Signaling Technology, Danvers, MA, USA, catalog number: 4842) or β-actin (Mouse anti-β-Actin antibody, Sigma-Aldrich, Saint-Louis, MO, USA, catalog number: A1978), at a 1:1000 dilution. The membrane was washed three times with PBS containing 0.1% Tween 20 and incubated with the appropriate horseradish peroxidase-conjugated secondary antibodies (Goat anti-Rabbit or Goat anti-Mouse, catalog numbers: 172-1019 and 172-1011, respectively) for one hour, at room temperature. After washing, revelation was performed using enhanced chemiluminescence reagent (Amersham, catalog number: RPN2232) and acquired with a Fusion Fx Spectra imager (Vilber, Marne-la-Vallée, France). Spot quantification from Western blot images was performed using the Fusion Fx Spectra software.

### 2.8. Statistical Analysis

Data are expressed as mean ± standard error (SEM) from at least four independent experiments. Statistical analysis was performed using one-way ANOVA followed by Bonferroni correction for multiple comparisons with GraphPad Prism 5 software (San Diego, CA, USA). Statistical significance was set at the 0.05 probability level.

## 3. Results

We first evaluated the potential cytotoxicity of ONNV, PIC, IL-β and CPT-11 on HSF. Basal levels of LDH (<10%) were detected in the supernatant of control cells. ONNV at MOI 0.1 and 1, but not PIC at 1 µg/mL and IL-1β at 10 ng/mL, had cytotoxic effects on HSF evidenced by higher levels of LDH release (21% and 45%, respectively) compared to control cells (Figure 1a). The expected cytotoxicity induced by ONNV was accompanied by an alteration of cell morphology (Appendix A), the detection of viral RNA (E1 and nsP2 genes) in HSF (Appendix A) and a decrease in MTT metabolism (Appendix A). Concentrations of CPT-11 ranging from 0.15 to 15 µM and covering the human plasma therapeutic range of 0.5 to 15 µM to treat cancer [20,21] were not cytotoxic for HSF (Figure 1b). We thus selected the highest concentration of 15 µM to study the potential modulatory effects of CPT-11 on the antiviral and pro-inflammatory responses of HSF exposed to ONNV, PIC and IL-1β. Of further note, this concentration of CPT-11 did not alter cell viability measured by the MTT assay (Appendix A), as well as cell morphology (Appendix A).

As featured in Figure 2a, ISG15 gene expression was upregulated in HSF after 24 h of treatment with PIC and ONNV. CPT-11 did not significantly change ISG15 expression in control cells but increased it by 2-fold in PIC- but not in ONNV-stimulated cells (Figure 2b). We thought to determine whether CPT-11 had any effect on ONNV replication following cells’ treatment with PIC. As shown in Appendix A, a higher ISG15 expression was measured in HSF exposed to ONNV and after a 24 h co-stimulation with PIC and CPT-11 (compared to the stimulation with PIC alone). ONNV replication (monitored by RT-qPCR experiments for E1 and nsP2) was dramatically reduced in HSF previously treated with PIC (63-fold decrease and 15-fold decrease of E1 and nsP2 mRNA levels, respectively). This was probably due to high ISG levels initially induced by PIC. CPT-11 did not interfere with this major effect (Appendix A).

Subsequently, we evaluated the expression of genes related to the antiviral response, including genes encoding for MDA5 (a nucleic acid sensor that recognizes viral long dsRNA mimicked here by PIC and that contributes to type I interferon expression [22]), IFN-β, STAT1 and STAT2 (two components of the transcription factor ISGF3 involved in ISGs production). RT-qPCR analysis revealed an upregulation of MDA5 and IFN-β expression in HSF cultured in the presence of ONNV and PIC (Figure 3a,b). However, CPT-11 did not influence the expression of MDA5 and IFN-β in both control and stimulated cells. STAT1 and STAT2 expression were significantly increased in cells exposed to PIC (by 4-fold and 3-fold, respectively, compared to control cells) but not in cells exposed to ONNV (Figure 3c,d). CPT-11 significantly amplified STAT1 and STAT2 expression by 1.5-fold only in PIC-treated cells (Figure 3c,d). Overall, these results suggest that CPT-11 positively regulates ISG15 expression by improving type I interferon signaling rather than type I interferon expression itself in HSF stimulated with long dsRNA but not with whole alphaviruses.

We also analyzed the expression of the critical tumor suppressor gene p53 upon CPT-11 treatment. Indeed, the ISG15 gene is known to be a p53-induced gene [23]. As shown in Figure 3e, ONNV and PIC significantly reduced p53 mRNA levels in HSF by 2.5-fold and 1.5-fold, respectively. PIC-induced ISG-15 up-regulation is thus not associated to p53 expression. CPT-11 increases p53 expression in control cells and in cells treated with ONNV but not in a significant manner. In PIC-treated cells, this expression was significantly higher in the presence of CPT-11 (2-fold increase).

Other ISGs were studied in cells simultaneously exposed to PIC and CPT-11. PIC-stimulated HSF displayed higher levels of ISG54 and 2′-5′-oligoadenylate synthetase 1 (OAS1) mRNA and lower levels of protein kinase R (PKR) mRNA compared to control cells (Appendix A). Similar to the ISG15 RT-qPCR results, CPT-11 was responsible for an elevation of ISG54, OAS1 and PKR expression during PIC exposure, further confirming the role of this drug in the enhancement of antiviral gene expression.

We next investigated the potential effects of CPT-11 on the pro-inflammatory response of HSF cultured in the presence of ONNV and PIC. This response was also studied in HSF treated with IL-1β, a cytokine involved in the pathogenesis of chronic and severe arthritis/arthralgia following alphaviral infection [24,25] and well-characterized to induce the expression of pro-inflammatory mediators, such as CXCL8, IL-6 and COX2, in synovial fibroblasts from patients with rheumatoid arthritis [26,27]. IL-1β was the most potent inducer of CXCL8, IL-6 and COX2 gene expression at 6 h of treatment compared to ONNV and PIC (Figure 4a–c). For instance, CXCL8 and COX2 expression levels were 60-fold and 20-fold higher, respectively, in IL-1β- than in PIC-stimulated HSF (Figure 4a,c). A reduced expression of CXCL8, IL-6 and COX2 was observed over time (at 24 h) upon IL-1β treatment. ONNV and PIC modestly but significantly increased IL-6 and COX2 expression at 24 h of treatment (Figure 4b,c). In unstimulated cells, CPT-11 significantly increased CXCL8 expression by 2-fold, while no effect of the drug occurred during PIC treatment and a 1.5-fold decreased expression was measured in IL-1β conditions (Figure 4d). CPT-11 did not influence CXCL8 secretion in control cells, suggesting that the increase in mRNA levels previously detected was not sufficient to generate a change at the protein level. Confirming RT-qPCR data, IL-1β-induced CXCL8 secretion (72 ± 2 ng/mL versus 0.04 ± 0.02 ng/mL in control condition) was significantly decreased up to 52 ± 3 ng/mL by CPT-11 (Figure 5a). Of note, PIC-induced CXCL8 secretion (0.6 ± 0.1 ng/mL versus 0.04 ± 0.02 ng/mL in control condition) was not modified by CPT-11 (data not shown). In contrast to CXCL8 results, CPT-11 had no effect on IL-6 gene expression in both unstimulated and stimulated cells (Figure 4e). Unexpectedly, ELISA revealed that IL-1β-induced IL-6 secretion (112 ± 19 ng/mL versus 0.2 ± 0.03 ng/mL in control condition) was significantly reduced at 59 ± 11 ng/mL by CPT-11 (Figure 5b). This discrepancy with RT-qPCR analysis suggests that CPT-11 regulates not only transcriptional, but also post-transcriptional processes. Regarding COX-2, its expression in control cells was relatively low, although a 4-fold significant increase was detected after 24 h of stimulation with CPT-11. No major effect of the drug was observed upon PIC and ONNV treatments, whereas a 1.5-fold decreased gene expression was measured in IL-1β conditions (Figure 4f). At the protein level, COX-2 was barely detectable by Western blot in control cells cultured with or without CPT-11. As expected, IL-1β-induced COX-2 production was dampened by CPT-11 **(**Figure 5c,d). Our results indicate that this CPT analogue displays anti-inflammatory properties in stimulated HSF. This effect was not limited to the IL-1β treatment since, in the presence of TNF-α (another pro-inflammatory cytokine present during the chronic phase of alphavirus disease [28]), CPT-11 also reduced CXCL8 mRNA and protein levels by 1.6-fold and 3-fold, respectively (Appendix A). Of note, COX-2 expression was not significantly upregulated by TNF-α (data not shown).

As the antagonistic pathways NF-κB and p53 predominantly modulate inflammation in synovial fibroblasts in the context of rheumatoid arthritis [26,29], RT-qPCR experiments were carried out to assess the expression of these two transcription factors. IL-1β markedly increased NF-κB1 gene expression (by 8-fold) at 6 h of treatment and this effect was completely blunted over time (at 24 h). CPT-11 did not modulate NF-κB1 expression in control cells but down-regulated it by 1.5-fold and in a significant manner in cells exposed to IL-1β (Figure 6a). The expression level of p53 was reduced by 1.6-fold and 3-fold in IL-1β-stimulated cells compared to control cells, at 6 and 24 h of treatment, respectively. IL-1β-induced decrease in p53 expression was significantly rescued by CPT-11 (2.4-fold increase) at 24 h (Figure 6b). These results corroborate the anti-inflammatory effects of CPT-11 on HSF cultured in the presence of IL-1β.

## 4. Discussion

Infection with alphaviruses is characterized by a viremia usually lasting 5–7 days and following a 2–6-day incubation period in adults. Acute disease is associated with a strong type I interferon response leading to ISG expression. While many patients with acute alphavirus (CHIKV) infection recover, 1.6–57% of them develop chronic arthritic disease that remains unresolved for up to several years [30,31]. The molecular mechanisms involved in the transition from acute to chronic pathology are still unclear. Persistence of viral antigens in synovial tissue and prolonged inflammation are thought to contribute to disease progression. Of critical note, viral dsRNA was detected in the joint of a human clinical case report, 18 months post-infection by CHIKV [32]. Moreover, the presence of arthritogenic dsRNA was confirmed in the synovial fluid from chronic rheumatoid arthritis patients with an erosive disease course [33]. The majority of all viruses produce dsRNA during their replication in host cells. In our model, PIC was thus applied to synovial fibroblasts to mimic, in general, the presence of dsRNA in the joint during the chronic phase of viral arthritic disease. There is currently no treatment or vaccine specific for alphavirus infection and, moreover, a previous investigation of our group revealed that the disease-modifying anti-rheumatic drug methotrexate had no effect on the antiviral and pro-inflammatory response of synovial fibroblasts exposed to PIC [34]. The originality of the present study was to evaluate the repositioning of the anti-cancer molecule CPT-11 (a semisynthetic analogue of camptothecin) in the context of viral arthritis. At the non-toxic concentration of 10 µg/mL (15 µM), we found that CPT-11 enhances antiviral gene expression (ISGs) in HSF treated with PIC (but not with ONNV). PIC is a synthetic analogue of viral double-stranded RNA, while alphavirus, in addition to be composed of RNA, also contains many other molecules such as nonstructural proteins (nsP). Among these proteins, nsP2 and nsP3 were found to inhibit host antiviral pathways. For instance, nsP2 decreases ISG expression by promoting transcriptional shutoff by degradation of RNA polymerase subunit RPB1 and by blocking the IFN-induced JAK-STAT pathway [35]. This may explain why mRNA levels of ISG15, STAT1 and STAT2 were not increased by CPT-11 after ONNV stimulation.

The enhanced ISG expression mediated by CPT-11 upon PIC treatment was associated with increased gene expression of STAT1 and STAT2, described to be involved in ISG expression. In line with our results, higher levels of ISG15 were evidenced in human fibrosarcoma cells in response to camptothecin, as well as the requirement of p53 in this mechanism [36]. In their genome-wide transcriptional analysis, Veloso and collaborators showed that interferon regulatory factors (IRFs), including IRF1, IRF3 and IRF9, were upregulated in human fibroblasts after 7 µg/mL CPT exposure (16). IRF1 and IRF3 are positive regulators of type I interferon gene induction [37,38]. Although we did not measure the expression of these two transcription factors in our model, we found no variation of IFN-β mRNA levels upon CPT-11 treatment. Upregulation of ISG expression in HSF could be linked to IRF1-stimulated p53 activity, as demonstrated in the human colon cancer cell line HCT116 [39], and/or p53 ISGylation that forms a positive feedback loop for its transcriptional activation [40], and/or increased ISGF3 production dependent on IRF9, STAT1 and STAT2 intracellular amounts.

Regarding inflammation, data about CPT and its derivatives are still unclear. Indeed, it has been demonstrated that CPT and CPT-11 induce activation of the NF-κB pathway in a variety of human carcinoma cell lines, a mechanism thought to contribute to resistance to chemotherapy [41,42,43]. In addition, SN38, the active metabolite of CPT-11, has been shown to increase CXCL8 secretion by enhancing the phosphorylation of MAP kinases in HCT8 cells [44]. By contrast, Riedlinger et al. evidenced that topoisomerase I inhibitors interfere with cytokine-induced and NF-κB p65-mediated inflammatory gene expression in a variety of cell lines [45]. In human fibroblasts, CPT was found to increase the expression of NFKBIE and NFKBIB genes (encoding for the NF-κB inhibitors IκB-ε and IκB-β, respectively) and decrease the expression of the IKBKB gene (encoding for the NF-κB activator IKK-β) [16]. In support to these latest findings, our data revealed that the expression of NFKB1 gene (encoding for the p105 subunit, precursor of the p50 subunit) was down-regulated in HSF co-treated with IL-1β and CPT-11. This result correlated with reduced CXCL8 and COX-2 mRNA and protein levels, whereas the secretion of IL-6 (a biomarker, among others, of arthralgia/arthritis persistence and severity following alphaviral infection [28,46]) appeared to be modulated at the post-transcriptional level. In this sense, post-transcriptional-level interactions between NF-κB and the Lin28/let-7 pathway were recently described in the review written by Mills IV et al. and showed that NF-κB mediates Lin28b expression, an inhibitor of Let-7 miRNA family members, themselves known to reduce IL-6 production by interfering with the 3′UTR region of its mRNA [47]. This review also evidences many predicted interactions between let-7 miRNA and mRNA related to NF-κB subunits, including, among others, interactions that repress p105 mRNA. Interestingly, several natural compounds (polyphenols and indole alkaloids) have been documented to intensify let-7 expression [48,49]. Further investigations are required to determine whether CPT and its analogues are able to modulate the NF-κB/Lin28/Let-7 pathway, as new miRNA-based therapies could be developed in the context of viral arthritis, but also other inflammatory diseases, such as autoimmune diseases, type 2 diabetes and obesity. Complementary studies should be also performed on the tumor suppressor gene p53. We found that the IL-1β-induced decrease in p53 expression was abolished by CPT-11, a result consistent with reduced CXCL8, IL-6 and COX-2 levels. Indeed, p53 and NF-κB function as mutual inhibitors. Zhang and collaborators established that p53 deficiency increased the phosphorylation rate of IkB-α and MAP kinases in fibroblast-like synoviocytes treated with IL-1β, leading to higher IL-6 secretion [29]. Of important note, the dual-specificity phosphatase 5 (DUSP5) gene, a direct target of p53, is upregulated in human fibroblasts, after CPT exposure [16]. DUSP5 is known to be induced in response to MAP kinase activation [50,51] and acts as a negative feedback regulator of MAP kinases and NF-κB signaling pathways in a phosphatase activity-dependent or -independent manner [52,53]. Although we did not analyze DUSP5 and MAP kinase expression in our model, it is likely that the level of HSF stimulation in the presence of IL-1β is sufficiently high for DUSP5 to be expressed and to act by inhibiting NF-κB and MAP kinase pathways. 

In conclusion, CPT-11 appears to be an interesting molecule, particularly in the context of chronic and severe arthritis due to viral infection. Indeed, this drug acts both as a potentiator of the antiviral response of HSF exposed to PIC and as an inhibitor of inflammation after IL-1β exposure. Further studies are warranted to address whether CPT-11 is able to modulate these responses in T cells and macrophages (also present in the synovial tissue of patients with infectious chronic inflammatory rheumatisms). This work also highlights that modulators of NF-κB and p53 could be therapeutically useful for the treatment of arthritis. Currently, the most common forms of administration of CPT-11 in patients are 30- or 90-min intravenous infusions of 125 mg/m^2^ given weekly for 4 of every 6 weeks (North America), or 350 mg/m^2^ given every 3 weeks (Europe) [54]. Although CPT-11 was not toxic for cultured HSF over the entire human therapeutic range, the molecule is often associated with intestinal mucosal injury and clinically significant diarrhea [55]. To reduce the CPT-11-linked systemic side effect, co-therapy with polyphenols could be envisaged. For instance, the protective effect of curcumin against irinotecan-induced intestinal mucosal injury was evidenced in nude mice [56]. Another strategy would be the local, controlled delivery of CPT-11 (in the case of arthritis, in the articular synovium) loaded in polymers, such as ethylene-vinyl acetate co-polymer [57].

## Figures and Tables

**Figure 1 cells-10-01431-f001:**
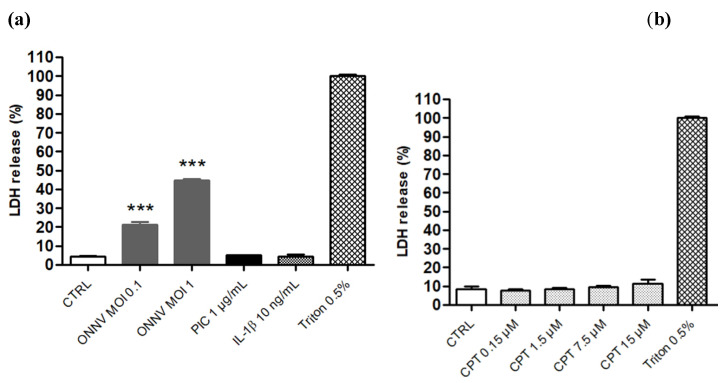
ONNV is cytotoxic for HSF. LDH activity was measured in culture supernatant 24 h after cell treatment with (**a**) ONNV at MOI 0.1 and 1, PIC at 1 µg/mL, IL-1β at 10 ng/mL, and (**b**) CPT-11 at 0.15 to 15 µM. The percentage of cytotoxicity was calculated from the maximum LDH release following Triton X-100 exposure. Results are expressed as mean ± SEM of four independent experiments. Statistical significance is indicated compared to control (CTRL), as follows: *p*-value < 0.001 (***).

**Figure 2 cells-10-01431-f002:**
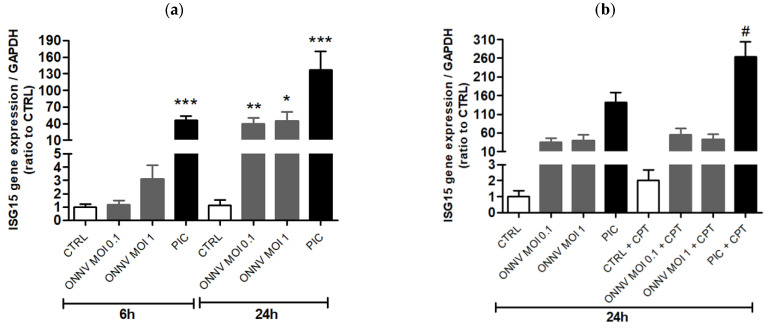
CPT enhances ISG15 gene expression in PIC-stimulated HSF. ISG15 mRNA levels from HSF treated with (**a**) ONNV (MOI 0.1 and 1) and PIC (1 µg/mL), and (**b**) co-treated with CPT-11 (15 µM), for 6 and 24 h, were evaluated by RT-qPCR. Results are expressed as mean ± SEM of four independent experiments. Statistical significance is indicated as follows: *p*-value < 0.05 (*), *p*-value < 0.01 (**), *p*-value < 0.001 (***) compared to control at the corresponding time of treatment; *p*-value < 0.05 (#) compared to the corresponding treatment without CPT-11.

**Figure 3 cells-10-01431-f003:**
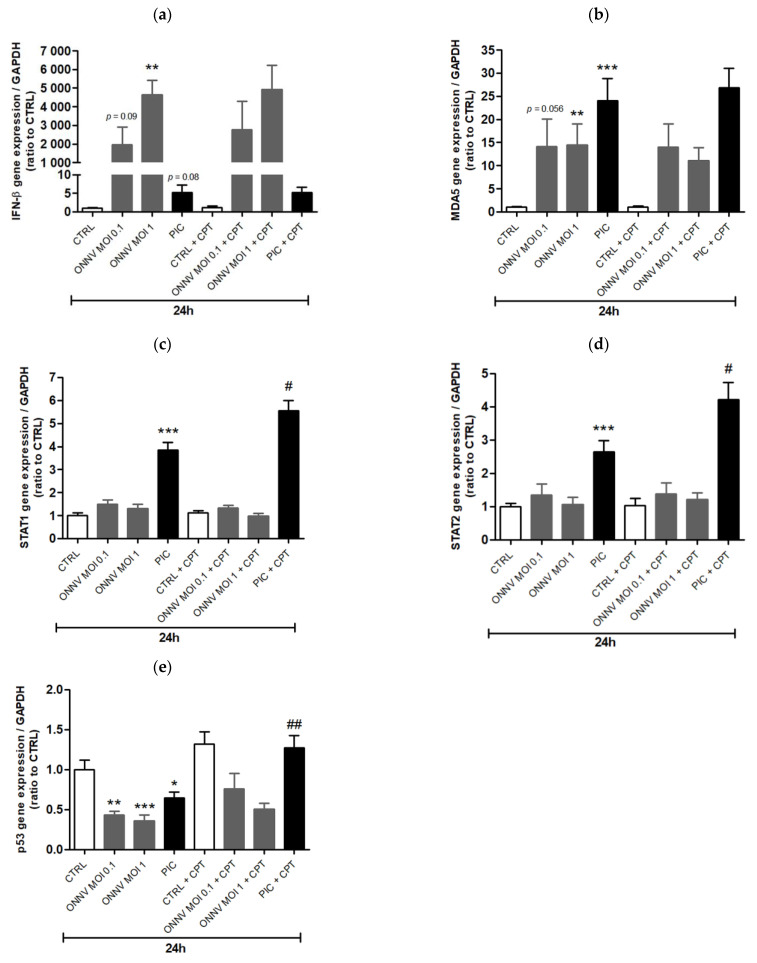
CPT-11 enhances STAT1, STAT2 and p53 (but not type I interferon and MDA5) gene expression in PIC-stimulated HSF. (**a**) IFN-β, (**b**) MDA5, (**c**) STAT1, (**d**) STAT2 and (**e**) p53 mRNA levels from HSF treated with ONNV and PIC and co-treated with CPT-11, for 24 h, were evaluated by RT-qPCR. Results are expressed as mean ± SEM of four independent experiments. Statistical significance is indicated as follows: *p*-value < 0.05 (*), *p*-value < 0.01 (**), *p*-value < 0.001 (***) compared to control at the corresponding time of treatment; *p*-value < 0.05 (#), *p*-value < 0.01 (##) compared to the corresponding treatment without CPT-11.

**Figure 4 cells-10-01431-f004:**
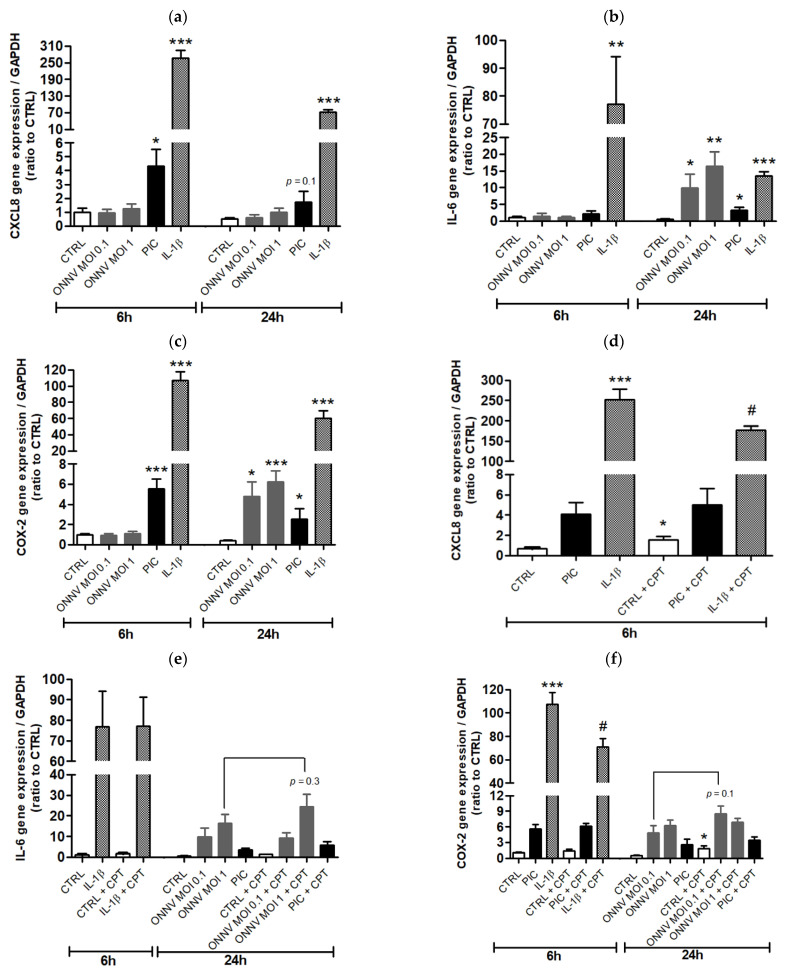
CPT-11 reduces IL-1β-induced CXCL8 and COX-2 (but not IL-6) expression in HSF. (**a**) CXCL8, (**b**) IL-6 and (**c**) COX-2 mRNA levels from HSF treated with ONNV, PIC and IL-1β, for 6 and 24 h, were evaluated by RT-qPCR. (**d**) CXCL8, (**e**) IL-6 and (**f**) COX-2 mRNA levels were next assessed in HSF treated in the presence of CPT-11. Results are expressed as mean ± SEM of four independent experiments. Statistical significance is indicated as follows: *p*-value < 0.05 (*), *p*-value < 0.01 (**), *p*-value < 0.001 (***) compared to control at the corresponding time of treatment; *p*-value < 0.05 (#) compared to the corresponding treatment without CPT-11.

**Figure 5 cells-10-01431-f005:**
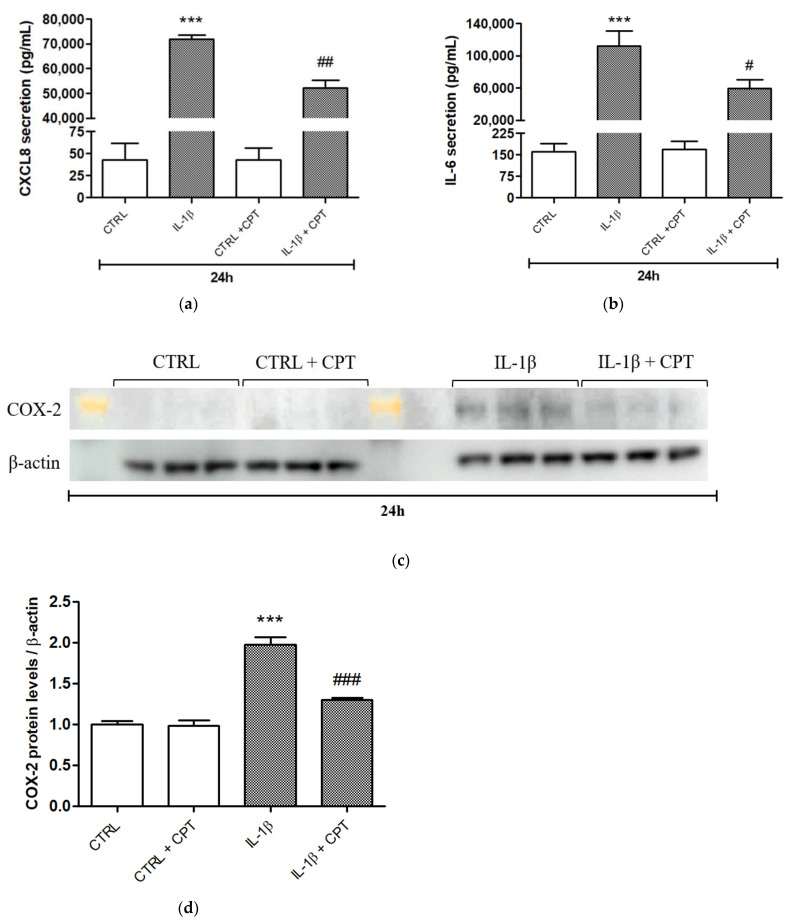
CPT-11 reduces IL-1β-induced pro-inflammatory mediators’ expression at the protein level, in HSF. (**a**) CXCL8 and (**b**) IL-6 secretion from HSF treated with IL-1β and co-treated with CPT-11, for 24 h, were evaluated by ELISA in cell supernatants. (**c**) COX-2 intracellular protein levels were measured by Western blot (n = 3). The yellow mark indicates a molecular weight of 76 kDa (from Amersham ECL Full-Range Rainbow Molecular Weight Markers, reference RPN800E). (**d**) Signal intensities from Western blot were quantified (arbitrary units). Quantitative results are expressed as mean ± SEM. Statistical significance is indicated as follows: *p*-value < 0.001 (***) compared to control at the corresponding time of treatment; *p*-value < 0.05 (#), *p*-value < 0.01 (##), *p*-value < 0.001 (###) compared to the corresponding treatment without CPT-11.

**Figure 6 cells-10-01431-f006:**
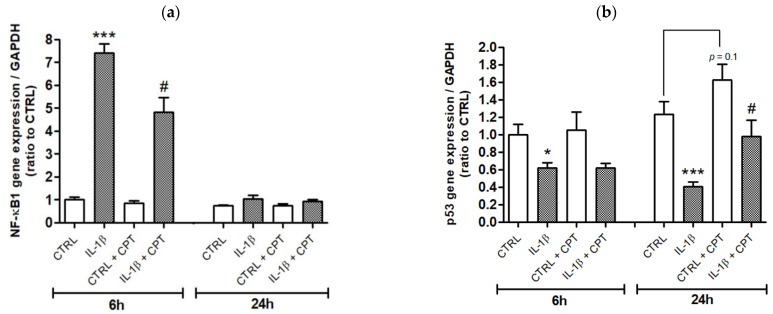
CPT-11 down-regulates IL-1β-induced NF-κB1 expression and upregulates an IL-1β-induced decrease in p53 expression in HSF. (**a**) NF-κB1 and (**b**) p53 mRNA levels from HSF treated with IL-1β and co-treated with CPT-11, for 6 and 24 h, were evaluated by RT-qPCR. Results are expressed as mean ± SEM of four independent experiments. Statistical significance is indicated as follows: *p*-value < 0.05 (*), *p*-value < 0.001 (***) compared to control at the corresponding time of treatment; *p*-value < 0.05 (#) compared to the corresponding treatment without CPT-11.

**Table 1 cells-10-01431-t001:** List of primers used for RT-qPCR.

Gene Name	Sequence (5′ → 3′)
GAPDH_F (Forward)	TGTTCGTCATGGGTGTGAAC
GAPDH_R (Reverse)	GCATGGACTGTGGTCATGAG
E1_F	CACCGTCCCCGTACGTAAAA
E1_R	GGCTCTGTAGGCTGATGCAA
nsP2_F	GCGGAGCAGGTAAAAACGTG
nsP2_R	TAGAACACGCCCGTCGTATG
ISG15_F	AGATCACCCAGAAGATCGGC
ISG15_R	GAGGTTCGTCGCATTTGTCC
IFN-β_F	GTTCGTGTTGTCAACATGACCAA
IFN-β_R	TCAATTGCCACAGGAGCTTCT
MDA5_F	CTGTTTACATTGCCAAGGATC
MDA5_R	ACACCAGCATCTTCTCCATTT
STAT1_F	TGGTGAAATTGCAAGAGCTG
STAT1_R	AGAGGTCGTCTCGAGGTCAA
p53_F	GAAGAGAATCTCCGCAAGAAAGG
p53_R	TCCATCCAGTGGTTTCTTCTTTG
ISG54_F	CTGGTCACCTGGGGAAACTA
ISG54_R	GAGCCTTCTCAAAGCACACC
OAS1_F	CATGCAAATCAACCATGCCA
OAS1_R	ACAACCAGGTCAGCGTCAGATC
PKR_F	GTGATGCAGCTCACAATGCT
PKR_R	GGCACTGTAAAATGGGTGCT
CXCL8_F	CAGAGACAGCAGAGCACACA
CXCL8_R	GGCAAAACTGCACCTTCACA
IL-6_F	TACAGGGAGAGGGAGCGATAA
IL-6_R	TGGACCGAAGGCGCTTGT
COX-2_F	TGGCTACAAAAGCTGGGAAG
COX-2_R	GGGGATCAGGGATGAACTTT
NFKB1_F	CCGGCCCGCCTGAATCATTCTC
NFKB1_R	CAGGTGGCGACCGTGATACCT

## Data Availability

The data presented in this study are available in article.

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
