# Peer review of "Irinotecan (CPT-11) Canonical Anti-Cancer Drug Can also Modulate Antiviral and Pro-Inflammatory Responses of Primary Human Synovial Fibroblasts"

_cells, 2021, doi:10.3390/cells10061431_

Round 1

Reviewer 1 Report

In this study the authors wanted to evaluate the possible effect of the anti- cancer molecule irinotecan as a potential modulator of the antiviral and inflammatory responses of primary human synovial fibroblasts (HSF), the main stromal cells of the joint synovium.

The authors extensively and satisfactory described the effect of irinotecan, while also evaluating its possible toxic effect on HSF cells.

-Are the authors sure that, even if the toxicity, the morphological damage is not present?

Did the authors prove the long-term effect?

Do cells not acquire toxicity in the long term? don't they become resistant?

Reviewer 2 Report

In the manuscript “Irinotecan (CPT-11) canonical anti-cancer drug can also modulate antiviral and pro-inflammatory responses of primary human synovial fibroblasts” authors main demonstrated that CPT-11 potentiates the interferon stimulated genes (ISGs) expression induced by PIC (a mimetic dsRNA) and reduce the secretion of IL-1beta-induced pro-inflammatory mediators. However, despite significant effect of CPT-1 1, the conduction of the study was based on a direct correlation with the chronic stage of arthritogenic alphavirus infections, mainly of the chikungunya virus. However, experimental design brings only an assessment of the effects of CPT-11 treatment in the context of acute stimulation with dsRNA and IL-1beta . Thus, several statements and interpretations of the results were made improperly along the study. In addition, there are some data inconsistences and speculative conclusions were made.  Specific considerations regarding the study are listed below.

  1. At the present work, human synovial fibroblast (HSF) was treated with polyinosinic-polycytidylic acid (PIC), a synthetic double-strand RNA, in order to mimic chronic infectious settings of alphavirus. The synthesis of dsRNA is a common step during RNA virus replication in host cells. Treatment of target cells with dsRNA alone is not a specific pathogen-associated molecular patternand also does not make it a chronic model of alphavirus stimulated cells.  In the present study, the effects of the CPT-11 treatment were evaluated during an acute stimulus of dsRNA, since HSF was co-treated with PIC and CPT-11 together. It does not make sense since the objective was to corelates the impact of CPT-11 treatment in a chronic stimulated model. The pretreatment of HSF with the PIC would be more appropriate in this case. The same rationale could be considered for the treatment with IL-1beta.

  1. PIC stimulation result in an increase in IFN-stimulated genes 15 and others, however do not induce IFN-beta expression. In addition, the alternative p53-ISG15 induction was not responsible for ISGs increasing since p53 expression was reduced after PIC stimulation.

  1. Some mediators in the study are not only modulated in gene expression levels. Thus, the increase in gene expression could not result on its activation. For example, STAT1/2 phosphorylation and NF-Kb translocation needs to be addressed to confirm the result of CPT-11. Besides it, the CPT-11 do not modulate antiviral genes in the course of viral infection (at least during ONNV infection). Would this modulation result in a relevant antiviral effect when the viral proteins, which restrict the interferon response, are being synthetized in the cells? Which is the relevance of modulation of antiviral genes in the context of chronic infection only in the presence of dsRNA? A functional test that demonstrates that PIC treated cells are comparative more restrictive to viral infection after treatment with CPT-11 should be included.

  1. The synovial fibroblast, despite be involved in the pathogenesis of arthritogenic alphavirus, it has not been considering a cell for chikungunya virus persistence. Literature evidences points muscle tissue and synovial macrophages as the major sites for persistence. Thus, only IL-1beta could be considered as a chronic signal for HSF cells. Despite reduction in IL-1beta-stimulated pro-inflammatory mediators by CPT-11, another’s mediators would be present during chronic phase of alphavirus disease, limiting a specific conclusion about the impact of CPT-11 treatment in chikungunya virus induced inflammatory response. The interpretation and conclusion should be limited to CPT-11 is able to reduce the secretion of pro-inflammatory mediator induced by IL-1beta.

  1. The effects observed after PIC and IL-1 beta stimulus should be extrapolated only as a ligand for dsRNA and IL-1 receptors. Would the same effect of CPT-11 treatment be found for a stimulus with TLR7 agonist? Or others agonist of PRRs? Or others pro-inflammatory cytokine? This is a very interesting issue to be addressed in order to investigate if CPT-11 has a potential effect to be considered as a broad-spectrum treatment for infectious disease.

Reviewer 3 Report

In the manuscript, the authors have investigated the effect of irinotecan, an anti-cancer drug on antiviral and inflammatory response of primary human synovial fibroblasts. The irinotecan treatment in IL-1β stimulated HSF led to increased anti-viral response measured by increase in interferon-stimulated gene signature and reduction in pro-inflammatory cytokine gene signature. The acute and severe arthritis model used would have been more insightful if different duration or concentration/MOI of either ONNV virus or PIC would have been used to mimic the pathological condition. Nevertheless, the data on PIC and IL-1β stimulation of HSF in presence of irinotecan is very convincing and presents an interesting aspect of the manuscript.    

There are few comments that authors need to address before the manuscript is deemed suitable for publication.

  1. In the manuscript, many of the changes observed in different gene expression is not coherent with expression profile of HSFs exposed to PIC or IL-1β in the presence and absence of CPT-11 as compared to ONVV infection. The authors should discuss why this difference is observed? Does the ONVV infection at higher MOI or longer duration mimic the severe arthritis condition and is this response similar to stimulation with PIC in the presence or absence of CPT-11?
  2. The expression changes are measured by RT-qPCR by normalizing the data with reference housekeeping gene GAPDH. Since most of the data is based upon gene expression data, it is advisable to normalize the housekeeping data using 2 or more housekeeping genes as normalizing control. Was there some other analysis done to show that GAPDH is the most stable housekeeping gene in these conditions?
  3. In the statistical analysis in the methods sections, the authors have written that the statistical significance was calculated by one-way ANOVA (followed by a two-tailed student’s T test) for multiple comparisons. Can authors describe this test? What was the multiple correction test used for one-way ANOVA test?
  4. The blots should be quantified in Fig5c to highlight the differences.

Round 2

Reviewer 3 Report

I thank the authors who accepted my considerations and made substantial changes to the manuscript. In my opinion, the revised manuscript is now suitable for the publication.

Author Response

Thank you so much for your positive response to the paper.